# The Advantage of Growth Hormone Alone as an Adjuvant Therapy in Advanced Age and BMI ≥ 24 kg/m^2^ with In Vitro Fertilization Failure Due to Poor Embryo Quality

**DOI:** 10.3390/jcm12030955

**Published:** 2023-01-26

**Authors:** Shuyi Jiang, Lingjie Fu, Wei Zhang, Na Zuo, Wenzheng Guan, Hao Sun, Xiuxia Wang

**Affiliations:** 1Center of Reproductive Medicine, Shengjing Hospital of China Medical University, 36 SanHao Street, Shenyang 110004, China; 2Department of Clinical Epidemiology and Evidence-Based Medicine, the First Hospital of China Medical University, 155 Nanjing North Street, Shenyang 110001, China

**Keywords:** growth hormone, in vitro fertilization, poor embryo quality, advanced age, obesity

## Abstract

This study aimed to assess the effects of GH adjuvant therapy on the cumulative live birth rate in patients with poor embryo quality and to determine the characteristics of patients who are more responsive to GH. A retrospective cohort study was carried out in patients who have suffered from previous IVF failure due to poor embryonic development and underwent IVF with or without a 6-week pretreatment with GH in the subsequent cycle from January 2018 to December 2020. Clinical parameters including the cumulative live birth rate between the (−) GH and (+) GH groups were compared. Multivariate analysis was performed to ascertain associations between clinical parameters and cumulative live birth rate. Upon analysis of the clinical data from 236 IVF cycles, 84 patients received GH and 152 did not receive GH. In frozen embryo transfer cycles, compared with the (−) GH group, the implantation rate and live birth rate were significantly higher in the (+) GH group (*p* < 0.05). After adjusting for possible confounding factors, GH improved cumulative live birth per oocyte retrieval cycle by 1.96 folds (*p* = 0.032). Furthermore, when patients were subdivided based on age and BMI, a significant increase in the cumulative live birth rate was found in the (+) GH group of patients between 35 and 42 years old and BMI ≥ 24 kg/m^2^, respectively (*p* < 0.05). GH may increase the live birth rate in women who experienced IVF failure because of poor embryonic development, particularly in obese patients and women with advanced age.

## 1. Introduction

Embryo quality is the main factor affecting the success rate of in vitro fertilization (IVF) programs, as the implantation rate and live birth rate increase when embryos with a high score are transferred [1,2], which are significantly correlated with age, obesity, oxidative stress, and other factors [3]. Some patients experience repeated failure of IVF pregnancy, mainly due to poor embryo quality. Among all the factors influencing embryonic developmental competence, oocyte quality plays a significant role in determining the development of high-quality embryos from normal fertilization. Therefore, improving oocyte quality is one of the effective means to improve embryo quality. In recent years, some adjuvant therapies have been employed in the IVF field to improve oocyte quality, such as growth hormone (GH) adjuvant therapy.

GH is a natural polypeptide that is mainly secreted by the anterior pituitary gland, but is expressed in many types of cells and tissues, where it regulates their functions. Initially, GH was thought of as a regulator of longitudinal growth, but subsequently, it has become known for its multiple regulatory effects on lipid, glucose, and protein metabolisms and modulating immune function [4,5]. Furthermore, GH has an important role in maintaining the normal reproductive function. It was shown that GH receptors (GHRs) were found in oocytes, granulosa cells, and stromal cells of the human ovary [6]. GH directly binds to GHRs via the sequential phosphorylation of Janus kinase 2 (JAK2) and the signal transducer and activator of transcription 5 (STAT5) control downstream signaling pathways, such as RAS/RAF/ERK1/2 and PI3K/Akt/mTOR pathways [7,8], which are important intracellular signaling pathways to regulate oocytes and granulosa cells proliferation and development. GH may indirectly affect oocyte quality through activating the synthesis of insulin-like growth factor (IGF-1) [9,10], which is a mediator of the classical biological effects on cell growth, development, and cellular proliferation.

Until now, GH-related studies mainly focused on patients diagnosed with a poor ovarian response (POR) and some randomized controlled trials (RCTs) showed that GH addition improved the clinical pregnancy rate [11,12,13] and live birth rate [14,15], while other research did not see any increase in the clinical pregnancy rate or live birth rate [16,17]. Last year, a Cochrane review reported that the effectiveness of GH on the live birth rate in POR patients was still unclear [18]. From the view of GH’s biological effects on oocyte development, it is speculated that GH may be more advantageous in improving oocyte competence than in improving ovarian response.

Because of the difficulty in evaluating oocyte quality in patients undergoing IVF [19,20,21], the cumulative live birth rate was used to observe the clinical efficacy, which is the most critical and valuable indicator to evidence the clinical outcome [22,23]. In this study, we aimed to assess whether GH addition as an adjuvant therapy could increase the cumulative live birth rate in patients with poor embryo quality in the previous IVF cycles. Furthermore, our current investigation planned to explore the characteristics of patients, which are more responsive to GH, and to further clarify the effect and potential mechanism(s) of GH adjuvant therapy on reproductive function.

## 2. Materials and Methods

### 2.1. Study Design and Patient

This retrospective study analyzed the data from January 2018 to December 2020 in the First Department of the Reproductive Medical Center of Shengjing Hospital of China Medical University. A subset of patients who had at least one previous unsuccessful cycle of assisted reproductive technology (ART) with no less than five correctly fertilized eggs but with extremely low embryo quality was included for the analyses.

Low embryo quality was described as the embryos obtained being included in the worst prognosis category according to the following: (1) >70% of embryos were poor-quality embryos based on the morphologic criteria assessed by the Association for the Study of Reproductive Biology (ASEBIR) [24]; (2) no transplantable embryos were obtained owing to the absence of euploid blastocysts after preimplantation genetic testing for aneuploidy (PGT-A); and (3) embryos of poor quality according to morphokinetic criteria established [25] by the embryoscope time-lapse system.

The exclusion criteria were endometriosis, untreated hydrosalpinx, untreated endocrine disorders, other uterine disorders that affected embryo implantation, and azoospermia in male partners. To offset the bias of patients receiving multiple drugs during the same cycle, only cycles where no adjuvant therapy or GH-only was administered were included, implying that cycles with other adjuvant treatments including dehydroepiandrosterone (DHEA) and melatonin were excluded in the final analysis. A total of 84 cycles in the (+) GH group and 152 cycles in the (−) GH group were included, and a stepwise diagram of the experimental design is displayed in Figure 1.

The study involving human participants was approved by the Ethics Committee of the Shengjing Hospital of China Medical University (Study approval number: 2020PS007F). Informed consent was obtained from all subjects involved in the study.

### 2.2. Clinical Management

Based on the physiological function of the patient and the experience of the clinician, an appropriate controlled ovarian stimulation (COS) protocol was selected with the starting doses of follicle-stimulating hormone (FSH) and highly purified human menopausal gonadotrophin (hp-hMG) ranging from 150 IU to 375 IU. In the (+) GH group, 2 IU of GH (Jintropin, GeneScience Pharmaceuticals Co., Ltd., Changchun, China) was administered subcutaneously daily from day 2 of the preceding menstrual cycle until gonadotropin (Gn) was administered, and then 5 IU of GH was given daily until ovum pick-up (OPU) (according to our Institute protocol). On the other hand, for the (−) GH group, no adjuvant therapy was administered. Most of the patients received conventional GnRH antagonist protocol or long agonist protocol to induce ovulation, while others used luteal phase support protocol or mild ovulation stimulation protocol. Gonadotropin was administered every day until the trigger day. The human chorionic gonadotropin (hCG) (Ovidrel, Merck Serono GmbH, Darmstadt, Germany) at a dose of 0.25 mg or gonadotropin-releasing hormone (GnRH) agonist (Diphereline, Ipsen Pharma, Paris, France) at a dose of 0.2 mg was injected for the final trigger when at least three dominant follicles reached 18 mm in diameter. Oocyte retrieval was performed under ultrasonographic guidance, 36 to 37 h after triggering.

### 2.3. Fertilization, Embryo Culture, and Embryo Transfer

Fertilization occurred by conventional insemination or by intracytoplasmic sperm injection (ICSI), depending upon the quality of the semen. The embryo was cultured using a standard protocol and scored using the Peter scoring system for cleavaged embryos [26] and the Gardner scoring system for blastocysts [27]. Good and fair quality embryos were categorized according to the Association for the Study of Reproductive Biology (ASEBIR) [24]. The indications for the freeze-all approach were patients with high progesterone levels (progesterone level of >1.5 ng/mL on hCG trigger day), prevention of ovarian hyperstimulation syndrome (OHSS), the endometrial condition did not meet the embryo transfer criteria, patients with recurrent implantation failure and chromosome abnormality whose embryos underwent preimplantation genetic screening, personal wishes, and other factors. Although single embryo transfer is strongly recommended, patients who have suffered from previous IVF failure because of deficient-quality embryos can receive up to two embryos in both fresh and frozen embryo transfer cycles. The luteal phase was supported by vaginal micronized progesterone (Utrogestan, Besins health care, Paris, France) at 200 mg three times daily or 90 mg vaginal progesterone gel (Crinone, Fleet Laboratories Ltd., Watford, UK) once daily.

### 2.4. Outcome Assessments

The primary clinical outcome was the cumulative live birth rate per oocyte retrieval cycle and live birth was identified as a live delivery of at least one fetus after 28 weeks of gestation. The cumulative live birth rate per oocyte retrieval cycle was defined as live deliveries (at least one live birth) per IVF cycle including fresh embryo transfer and frozen-thawed embryo transfer. Multiple births were counted as the number of live deliveries. Serum β-hCG levels of >50 U/L after embryonic transplantation were used to confirm a chemical pregnancy. Clinical pregnancy was defined by presence of a gestational sac at 7 weeks of gestation. The miscarriage rate was identified as the pregnancy loss relative to the number of embryo-transfer cycles initiated.

### 2.5. Statistical Analysis

Data were expressed as median (interquartile range, IQR) for continuous variables or number (percentage) for categorical variables and were analyzed using SPSS version 22.0 (SPSS Inc., Chicago, IL, USA). The Kolmogorov–Smirnov test was used to test the normality of continuous variables and a two-tailed t-test (for normally distributed data) or the Mann–Whitney U-test (for skewed data) was used to compare the difference between the groups, where appropriate. The Chi-squared test was used to test the proportions of categorical variables. Logistic regression was performed to assess the independent contributions of individual confounding parameters on outcomes, including age, body mass index (BMI), anti-Müllerian hormone (AMH) concentration, basal FSH level, antral follicle count (AFC), and number of follicles with ≥16 mm diameter at the day of trigger, in addition to the quality and number of embryos. The unadjusted effect of GH administration on the outcome variables was also assessed. The effect of each variable was expressed as an OR with a 95% CI. Stepwise multiple logistic regression analyses were performed to investigate the relationship between GH adjuvant therapy and cumulative live birth using *p* < 0.10 of the likelihood ratio test for inclusion. A *p*-value of less than 0.05 was considered statistically significant.

## 3. Results

A total of 236 patients (84 in the GH group and 152 in the control group) were eligible for the analysis. Baseline demographic characteristics retrieved for the analysis are summarized in Table 1. Specific to the patient cohort, age, BMI, duration of infertility, and basal FSH level between the (+) GH and (−) GH groups showed no significant differences (*p* > 0.05). While AFC (14.00 versus 10.00, *p* = 0.001) and AMH levels (3.62 versus 2.85, *p* = 0.021) were significantly lower in the (+) GH group. From the lower AFC and AMH level, it was shown that the (+) GH group was at a disadvantage of the underlying ovarian reserve conditions.

Comparing the two groups, no significant differences were found in cycle characteristics and controlled ovarian hyperstimulation (COH) variables (Table 2) such as the number of follicles with ≥16 mm diameter at the day of trigger, endometrial thickness at the day of trigger, peak E2 at the day of trigger, number of oocytes, number of 2PN, and number of transferable embryos. The length of Gn stimulation (*p* = 0.001) and total dose of Gn used for stimulation (*p* = 0.002) were significantly higher in the (+) GH group as compared with the control group. In the (+) GH group, 13 patients decided to undergo preimplantation genetic testing (PGT) on their embryos and, in the (−) GH group, 25 patients had PGT.

Only 21 and 13 fresh embryos were transferred in the (−) GH group and (+) GH group, respectively (Table 3). The implantation rate, chemical pregnancy rate, clinical pregnancy rate, miscarriage rate, and live birth rate were comparable between the two groups in those respective cycles. More frozen–thawed embryo transfers were performed in the (−) GH group compared with the (+) GH group (122 versus 68, respectively), and the implantation rate (39.0% versus 58.5%, *p* = 0.003), chemical pregnancy rate (54.1% versus 70.6%, *p* = 0.026), and live birth rate (41.8% versus 58.8%, *p* = 0.024) were significantly higher in the (+) GH group compared with the control group. The clinical pregnancy rate (53.3% versus 64.7%, *p* = 0.127) and miscarriage rate (11.5% versus 5.9%, *p* = 0.207) were comparable between the two groups in frozen embryo transfer cycles.

Table 4 shows that no significant differences in the clinical pregnancy rate and live birth rate per transfer cycle were found between the two groups. Although the cumulative live birth rate per oocyte pick-up cycle in the (+) GH group was higher than that in the (−) GH group, no statistically significant difference was observed (38.2% versus 51.2%, *p* = 0.053). However, in frozen embryo transfer cycles, live birth rate (41.8% versus 58.8%, *p* = 0.024) was significantly higher in the (+) GH group.

Table 5 shows the calculated cumulative live birth per oocyte retrieval cycle OR for each individual variable. Only patient’s age, number of follicles with ≥16 mm diameter at the day of trigger, number of oocytes, number of 2PN, number of transferable embryos, and number of good quality embryos were significant predictors of the cumulative live birth per oocyte retrieval cycle. Additionally, univariate analysis showed that the administration of GH had marginal statistical significance (*p* = 0.054). When performing stepwise multiple logistic regression, using terms that satisfy the variable with a statistical significance of *p* < 0.10, only the patient’s age, number of transferable embryos, and presence of (+) GH were significant. Increasing age of the patient decreased the chance of cumulative live birth per oocyte retrieval cycle by about 13% per advancing year. Most importantly, following adjustment for the age of the patient, number of follicles with ≥16 mm diameter at the day of trigger, number of oocytes, number of 2PN, number of transferable embryos, and number of good quality embryos, parameters that were critical for outcomes and (+) GH significantly increased the chance of cumulative live birth per oocyte retrieval cycle by 1.96 folds (95% CI 1.06 to 3.64, *p* = 0.032).

When the patient data based on different ages were analyzed, it was found that the effect of GH was related to the patient’s age. Women aged between 35 and 42 years were 2.55 times more likely to achieve a cumulative live birth per oocyte retrieval cycle in the (+) GH cycles (95% CI, 1.05 to 6.24; *p* = 0.040) (Figure 2). On the other hand, GH administration did not increase the likelihood of cumulative live births in those younger than 35 years or older than 42 years old.

Moreover, when the clinical outcome was analyzed by group according to BMI, the cumulative live birth rate per oocyte retrieval cycle of patients who received GH with a BMI of ≥24 kg/m^2^ was increased 3.61 times (95% CI, 1.35 to 9.71; *p* = 0.010) relative to the (−) GH group (Figure 2). Although GH did not increase the likelihood of cumulative live birth rate per oocyte retrieval cycle significantly in the group with a BMI of less than 18.5 kg/m^2^, GH-administered cycles achieved more cumulative live births in each oocyte retrieval cycle.

## 4. Discussion

In this study, we investigated the effect of GH administration on IVF treatment in patients with a history of IVF failure due to poor embryo quality. It was found that GH was an independent factor that enhanced the cumulative live birth rate per oocyte retrieval cycle. Moreover, in the subgroup of patients aged between 35 and 42 years and the subgroup of patients with a BMI of more than 24 kg/m^2^, GH displayed a more advantageous effect.

To the best of our knowledge, this study was the first study to analyze the effectiveness of cotreatment with 6-week GH alone on cumulative live births in each oocyte retrieval cycle in patients who had a history of poor embryo quality in the previous IVF/ICSI cycle. Compared with the traditional use of clinical pregnancy rate and live birth rate to evaluate the success of embryo transfer, it is more appropriate to use the cumulative success rate per patient after one complete oocyte retrieval cycle, including all fresh and frozen–thawed embryo transfer cycles from one oocyte retrieval. By employing the definition of one complete IVF/ICSI cycle, the total reproductive potential of one oocyte retrieval provides an all-inclusive success rate, which is relevant and significant [23]. As a series of studies suggested that GH regulates ovarian steroidogenesis [28,29,30] and follicle development [31,32], it is better to include the total number of oocytes in an IVF/ICSI cycle to evaluate the efficacy of GH using the cumulative live birth rate.

Many of the clinical studies on the efficacy of GH focused on patients who were diagnosed with the POR, but it was still unclear whether GH could play a role in the patient cohort with a history of low embryo quality. These days, how to improve the poor embryo quality is one of the most challenging tasks in the field of IVF/ICSI. Therefore, the data from patients with previous IVF/ICSI failure mainly due to the low embryo quality (the diagnostic criteria of low embryo quality were modified according to Labarta et al. [33]) were solely collected, and it was found that GH improved cumulative live births in the follow-up oocyte retrieval cycle among that patient cohort. To the best of our knowledge, only two studies explored the effect of GH administration in patients undergoing IVF/ICSI with a previous history of low embryo quality. The first study, performed in 2019 by Liu et al. [34], concluded that GH adjuvant therapy increased the clinical pregnancy rate, but the live birth rate was not considered in the analysis. The other study assessed the effectiveness of GH in patients with IVF cycle failure and no top-quality embryos produced. Compared with our study, GH was administered using a short-term protocol and it was found that GH addition supported more live births in IVF/ICSI cycles, which was consistent with our results. Although current studies on GH effectiveness in patients with poor embryo quality are limited, the comprehensive results of all studies suggested that GH may be beneficial in improving clinical outcome. Based on both human and animal investigations, the GH adjuvant therapy may be beneficial because GH improved oocyte competence by increasing the number of FSHR [12] and mitochondrial DNA copy number of granulosa cells [14], alleviating oxidative stress [13], and promoting estradiol synthesis [16,28], thereby improving embryo quality and leading to the success of IVF/ICSI. 

In addition to evaluating the effect of GH adjuvant therapy in patients with a previous history of low embryo quality, the age of patients as an independent factor in successful IVF outcomes was also investigated. Furthermore, in patients aged between 35 and 42 years, GH administration substantially increased the probability of live births, which was consistent with the results reported by Keane et al. [35] and Cai et al. [36]. Collectively, these studies’ data showed an age-dependent effect of GH administration. Previous studies found that GH secretion was decreased in both humans [37] and mice models [38], which may have been related to reduced fertility in patients with advanced maternal age. It was also found that GH could reverse the depletion of ovarian reserve due to age and improve oocyte quality by decreasing apoptosis [38]. The changing role of women in society led to a delay in childbearing, which resulted in older women having children. Many studies focused on how to improve the pregnancy outcome in patients with advanced age, and our study suggested that GH as a natural anti-aging peptide may improve IVF pregnancy outcome in older women. On the other hand, GH application did not show an advantage in young patients, which may be because of the fact that GH secretion was not much deficient in patients younger than 35 years old [39].

In this study, we also found that patients in the subgroup with a BMI of ≥24 kg/m^2^ were more likely to achieve cumulative live birth in the (+) GH cycles. Some studies demonstrated that GH secretion was decreased after weight gain [40,41] and GH secretion was increased when there was weight loss in obesity [42]. Obese individuals had disrupted GH secretion [43] and GH replacement improved body composition in young men with abdominal obesity [44]. From this study, it was thought that, because of the low GH secretion levels in obese infertile patients, exogenous GH administration could increase the levels of GH and IGF-1, thereby promoting lipolysis and affecting fat volume and distribution [45]. As obesity itself is a key factor affecting fertility, when it improves, it can enhance reproductive function. To the best of our knowledge, this was the first study to report the effect of GH in obese infertile women. Although the underlying mechanism of obesity suppressing GH secretion is still unclear, our results suggested that GH could improve cumulative live birth in obese women, which may offer a new concept for further research on GH in the field of infertility.

A recent meta-analysis concluded that GH reduced the dose of Gn required (SMD, −1.05; 95% CI, −1.62 to −0.49) in patients with a poor ovarian response [46], and the same trend was also found in the current study. Additionally, the duration of Gn administration was significantly shorter, which was consistent with the results of Bassiouny et al. [16] and Ob’edkova et al. [47]. GH may assist controlled ovulation by gonadotrophins.

The implantation rate, chemical pregnancy rate, and live birth rate of frozen embryo transfer in our study were significantly higher in the (+) GH cycles; however, this conclusion was not achieved in the fresh embryo transfer cycles, which may have been because of the small sample size in the fresh embryo transfer cycles.

Investigators hypothesized that GH administration might lead to a greater number of oocytes and transferable embryos, and thus suggested that GH may exert an effect on oocyte quality [11,48]. However, in the present study, we found no alteration due to GH administration in the number of oocytes or quality of embryos when embryo quality was judged by morphological characteristics. It was also previously demonstrated that GH administration augmented the number of good grade embryos slightly but significantly in women with POR (*p* = 0.046) [49]. Nonetheless, we did not detect such evidence through morphological evaluation of embryos in this study, morphological features may not have fully reflected the developmental potential of embryos, and this result was consistent with the results portrayed by Keane et al. [35].

Recently, a Cochrane systematic review containing data from fourteen RCTs that analyzed the GH in patients with POR demonstrated that it was very ambiguous about the effectiveness of GH on the live birth rate in IVF patients [18]. The POR is mainly linked to increased maternal age and ovarian reserve; however, short-term use of GH hardly improves the ovarian reserve. Therefore, it may not be easy to obtain a definite therapeutic effect by adding GH to poor responders. On the other hand, compared with the included RCTs, we employed a six-week pretreatment GH regimen in this study, which was a better dosing regimen of GH to achieve the optimal therapeutic effect [13,35,36], mainly because GH begins to play a role from the initial stage of oocyte development.

Hormone exposure not only plays a beneficial role in the target organs, but also has side effects [50,51]. In GH-deficient adults, the common side effects of GH long-term administration were fluid retention, carpal tunnel syndrome, paresthesias, and worsening of glucose tolerance [52]. Although GH had a neurological effect in the brain, it did not increase the risk of central nervous system neoplasms associated with GH exposure [53]. Till now, it was unclear of the adverse effect of GH in normal responders undergoing IVF [18].

As it was a retrospective analysis, some limitations were seen in the current study, including patient selection bias. However, to offset this selection bias, all of the IVF cycles that satisfied the criteria within the study period were included. Furthermore, the patients who received GH were suggested by experienced clinicians, but they had to pay for this additional treatment, and affordability was not taken into consideration in the current study, which could be overcome by conducting placebo-controlled studies in the future. As the inclusion criteria of poor embryo quality were strictly adopted in this study, the number of patients was limited, and further expansion of the sample size was needed to confirm the conclusions.

## 5. Conclusions

In conclusion, the data from our current study on GH addition provided extra evidence to illustrate the potential advantages of GH administration in patients in the IVF field. GH may improve fertility through increasing the circulating and localized GH level in elderly patients with decreased GH secretion due to age and regulating lipid metabolism in obese women. Although this study possessed certain limitations, the data suggested that GH administration offered more cumulative live births in patients with the previous failure due to poor embryo quality, particularly in women aged between 35 and 42 years and with BMI of ≥24 kg/m^2^, which also presented a new idea as the suitable population for GH adjuvant therapy.

## Figures and Tables

**Figure 1 jcm-12-00955-f001:**
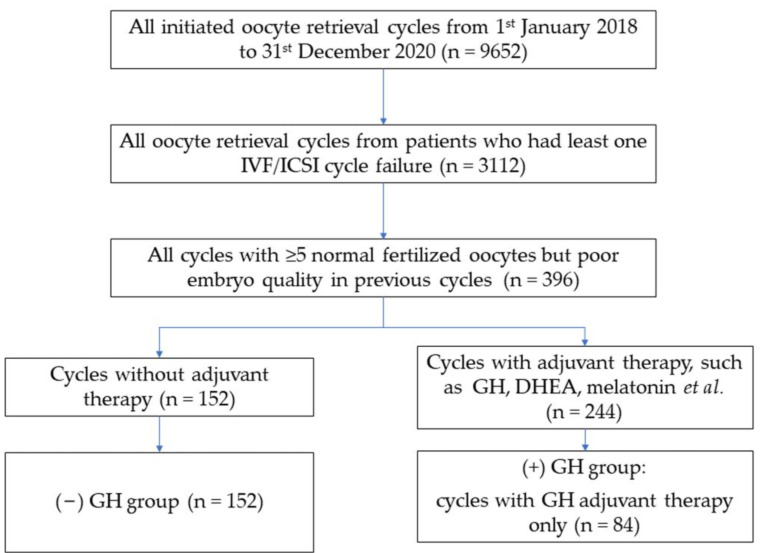
Flow diagram of data extraction. Data were extracted from the First Department of the Reproductive Medical Center of Shengjing Hospital of China Medical University. GH, growth hormone. DHEA, dehydroepiandrosterone.

**Figure 2 jcm-12-00955-f002:**
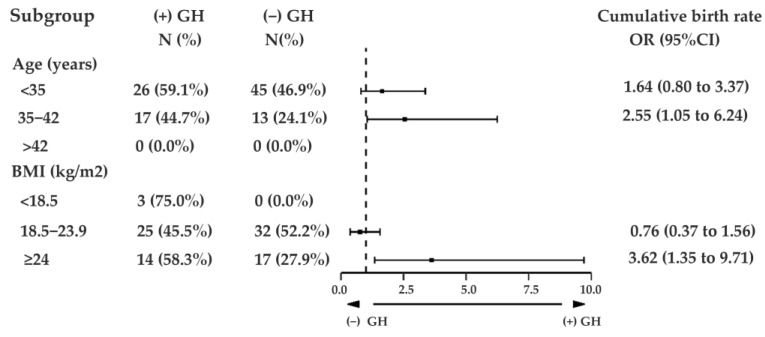
Effect of GH on cumulative live birth chance per oocyte retrieval cycle according to different groups of patient’s age and BMI. GH, growth hormone.

**Table 1 jcm-12-00955-t001:** Overview of baseline demographic data of (−) GH and (+) GH groups.

	(−) GH	(+) GH	*p*-Value
	n = 152	n = 84	
Age (year)	32.00 (29.00–36.00)	34.00 (30.25–37.00)	0.062
BMI (kg/m^2^)	23.40 (20.60–25.40)	22.00 (20.40–24.00)	0.070
Duration of infertility (years)	3.50 (2.33–5.73)	3.42 (2.08–5.42)	0.483
Cause of infertility			0.902
1—male factor	10 (6.58%)	7 (8.33%)	
2—anovulation	14 (9.21%)	6 (7.14%)	
3—tubal block	71 (46.71%)	36 (42.86%)	
4—chromosomal abnormality	8 (5.26%)	7 (8.33%)	
5—unexplained	28 (18.42%)	15 (17.86%)	
6—combined	21 (13.82%)	13 (15.48%)	
AFC (n)	14.00 (10.00–23.50)	10.00 (8.00–15.00)	0.001 *
Basal FSH (IU/L)	6.90 (5.61–8.08)	7.06 (6.08–8.32)	0.239
AMH (ng/mL)	3.62 (2.28–5.64)	2.85 (1.82–4.85)	0.021 *

Continuous variables are expressed as median (IQR); category variables are expressed as number (%); *: *p* < 0.05 indicates significant difference.

**Table 2 jcm-12-00955-t002:** Cycle characteristics and COH variables of the (−) GH and (+) GH groups.

	(−) GH	(+) GH	*p*-Value
Cycles conducted with antagonist protocol (n)	77 (50.65%)	46 (54.76%)	0.546
Cycles conducted with long protocol (n)	35 (23.03%)	20 (23.81%)	0.892
Cycles conducted with other protocols (n)	40 (26.32%)	18 (21.43%)	0.404
Cycles conducted PGT	25 (16.44%)	13 (15.48%)	0.846
Length of Gn stimulation (day)	9.00 (8.00–10.00)	9.00 (8.00–9.75)	0.001 *
Total dose of Gn stimulation (IU/mL)	2250.00 (1800.00–2775.00)	1950.00 (1575.00–2643.75)	0.002 *
No. of follicles ≥16 mm at the day of trigger (n)	8.00 (5.00–10.00)	7.00 (6.00–9.00)	0.694
Endometrial thickness at the day of trigger (mm)	10.00 (9.00–12.00)	11.00 (8.50–12.75)	0.455
Peak E2 at the day of trigger (pg/mL)	2964.00 (2069.25–5085.50)	3480.50 (1760.00–4852.50)	0.983
No. of oocytes (n)	12.00 (7.00–15.75)	11.00 (7.00–15.00)	0.747
Fertilization type			0.201
IVF (n)	71 (46.71%)	32 (38.10%)	
ICSI (n)	81 (53.29%)	52 (61.90%)	
No. of 2PN (n)	7.00 (4.00–10.75)	8.00 (6.00–10.00)	0.547
No. of transferable embryos (n)	6.00 (3.00–9.00)	6.00 (4.00–9.00)	0.265
No. of good and fair quality embryos (n)	2.00 (1.00–4.00)	2.00 (0.25–5.00)	0.449

Continuous variables are expressed as median (IQR); category variables are expressed as number (%); *: *p* < 0.05 indicates significant difference. Gn, gonadotropin.

**Table 3 jcm-12-00955-t003:** Reproductive outcomes in the study groups.

	(−) GH	(+) GH	*p*-Value
Fresh ET cycles (n)	21	13	
Fresh ET implantation rate, n (%)	10/42 (23.8)	3/24 (12.5)	0.266
Fresh ET chemical pregnancy rate, n (%)	10/21 (47.6)	3/13 (23.1)	0.601
Fresh ET clinical pregnancy rate, n (%)	8/21 (38.1)	3/13 (23.1)	0.363
Fresh ET miscarriage rate, n (%)	1/21 (4.8)	0/13 (0.0)	0.425
Fresh ET live birth rate, n (%)	7/21 (33.3)	3/13 (23.1)	0.524
Frozen ET cycles (n)	122	68	
Frozen ET implantation rate, n (%)	71/182 (39.0)	48/82 (58.5)	0.003 *
Frozen ET chemical pregnancy rate, n (%)	66/122 (54.1)	48/68 (70.6)	0.026 *
Frozen ET clinical pregnancy rate, n (%)	65/122 (53.3)	44/68 (64.7)	0.127
Frozen ET miscarriage rate, n (%)	14/122 (11.5)	4/68 (5.9)	0.207
Frozen ET live birth rate, n (%)	51/122(41.8)	40/68 (58.8)	0.024 *

*: *p* < 0.05 indicates significant difference.

**Table 4 jcm-12-00955-t004:** Cumulative clinical pregnancy and live birth rates in the study groups.

	(−) GH	(+) GH	*p*-Value
Clinical pregnancy			
Fresh cycle, n (%)	8/21 (38.1)	3/13 (23.1)	0.363
Frozen cycle, n (%)	65/122 (53.3)	44/68 (64.7)	0.127
Clinical pregnancy rate per transfer cycle	73/143 (51.0)	47/81 (58.0)	0.315
Live birth			
Fresh cycle, n (%)	7/21 (33.3)	3/13 (23.1)	0.524
Frozen cycle, n (%)	51/122 (41.8)	40/68 (58.8)	0.024 *
Cumulative live birth rate per transfer cycle	58/143 (40.6)	43/81 (53.1)	0.07
Cumulative live birth rate per oocyte retrieval cycle	58/152 (38.2)	43/84 (51.2)	0.053

*: *p* < 0.05 indicates significant difference.

**Table 5 jcm-12-00955-t005:** Association between clinical parameters and cumulative live birth rate.

	Cumulative Live Birth OR (95% CI)			
	Univariate Analysis	*p*-Value	Multivariate Analysis	*p*-Value
GH				
(−) GH	1.00	-	1.00	-
(+) GH	1.70 (0.99–2.91)	0.054	1.96 (1.06–3.64)	0.032 *
Age	0.88 (0.82–0.93)	<0.001	0.87 (0.81–0.93)	<0.001 *
BMI	0.94 (0.87–1.02)	0.115		
AMH	1.02 (0.95–1.11)	0.587		
FSH	1.00 (0.89–1.13)	0.988		
AFC	1.00 (0.95–1.05)	0.891		
No. of follicles ≥16 mm at the day of trigger (n)	1.12 (1.04–1.20)	0.004	1.08 (0.95–1.22)	0.221
Endometrial thickness at the day of trigger	1.01 (0.91–1.12)	0.830		
Number of oocytes	1.06 (1.02–1.11)	0.005	0.92 (0.83–1.03)	0.131
Number of 2PN	1.12 (1.05–1.19)	0.001	0.97 (0.80–1.17)	0.722
Number of transferable embryos	1.18 (1.10–1.27)	<0.001	1.24 (1.02–1.51)	0.033 *
Number of good quality embryos	1.20 (1.08–1.33)	0.001	1.03 (0.89–1.19)	0.720

*: *p* < 0.05 indicates significant difference.

## Data Availability

The datasets used and/or analyzed during the current study are available from the corresponding author upon reasonable request.

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
