# Peer review of "The Advantage of Growth Hormone Alone as an Adjuvant Therapy in Advanced Age and BMI ≥ 24 kg/m2 with In Vitro Fertilization Failure Due to Poor Embryo Quality"

_jcm, 2023, doi:10.3390/jcm12030955_

Round 1
Reviewer 1 Report
- Criteria for normal responder in introduction is necessary s
Comment 1
Sample age group did not meet the normal responder according to Posoidon criteria PMID: 28232864
Comment 2
Line 255 consider not to use the first study (Growth hormone dosage and protocol references PMID: 35641113, PMID: 31396161, PMID: 36452322,PMID: 32155012, PMID: 20457541)
Comment 3
Line 72 to 74 statement needs to provide references
(Oocyte quality references PMID:2701212, PMID: 35136962, PMID: 19667744, PMID: 34338482)
Comment 4
Line 301 Check spelling poorthe
Comment 5
Add significant P value cut point below every table
Comment 6
Explain P value 0.001 in Table 1, P value 0.024 in Table 4
Comment 7
Explain in discussion and conclusion for statement in introduction line 76- 78
Reviewer 2 Report
In this paper, Jiang and colleagues studied the effects of GH pretreatment on the cumulative live birth rate in normal responders with poor embryo quality. They comprised, in their study, patients underwent to IVF failure, and divided them into two groups: without or with a 6-week pretreatment with GH. They found that the implantation rate of frozen embryo and live birth rate were significantly higher in the (+) GH group. Interestingly, they also found an increase of the cumulative live birth rate patients between 35 and 42 years old and with a BMI ≥24 kg/m2.
The paper is interesting and may be of helpful for those working in the field. I therefore recommend for its publication in JCM prior these minor revisions:
- Figures 1 and 2 are a little blurred, and the used font is different from that of the text and tables;
- In the Materials and Methods section, please add the Ethical committee approvement code and whether patients produced an informed content
- In Table 2, the indicated p value of the parameter “length of Gn (should it be GH?) stimulation seems too much high, considering that in both groups the average is 9 days;
- In line 188, why just few fresh embryos were transferred?
- The GH supplementation that increased the birth rate in patients between 35 and 42 years old is very interesting, however, how can the authors explain the fact that in the younger women the GH treatment does not produce the same effect? I suggest to clarify this point.
Round 2
Reviewer 1 Report
Suggestion 1
Title change to The advantage of Growth hormone alone as an adjuvant therapy in advanced age and BMI ≥24 kg/m2 with In Vitro Fertilization failure due to poor embryo quality
Suggestion 2
line 362-364 GH may improve fertility through increasing circulating and localized GH level in elderly patients with decreased GH secretion due to age, and regulating lipid metabolism in obese women. move to Line 358
Line 361 for exploring change to as
Suggestion 3
recalculate P value (same number of days with impossible significant difference)
In Table 1 Length of Gn stimulation (day) 9.00 (8.00-10.00) 9.00 (8.00-9.75) P value 0.001*
Suggestion 4
Add (according to our Institute protocol) after the sentence in Line 116
Suggestion 5
Change GH adjuvant therapy in all GH therapy and GH treatment in the literature
Suggestion 6
add alone after GH in Line 255
Suggestion 7
Delete line 364 sentence (or) explain what kind of further studies
